# MUSEG: Reinforcing Video Temporal Understanding via Timestamp-Aware Multi-Segment Grounding

## Abstract

Video temporal understanding is crucial for multimodal large language models (MLLMs) to reason over events in videos. Despite recent advances in general video understanding, current MLLMs still struggle with fine-grained temporal reasoning. While reinforcement learning (RL) has been explored to address this issue recently, existing RL approaches remain limited in performance on time-sensitive tasks. In this work, we propose **MUSEG**, a novel RL-based method that enhances temporal understanding by introducing timestamp-aware multi-segment grounding. MUSEG enables MLLMs to align queries with multiple relevant video segments, promoting more comprehensive temporal reasoning. To facilitate effective learning, we design a customized RL training recipe with phased rewards that progressively guides the model toward temporally grounded reasoning. Extensive experiments on temporal grounding and time-sensitive video question answering (QA) tasks demonstrate that MUSEG significantly outperforms existing methods and generalizes well across diverse temporal understanding scenarios.

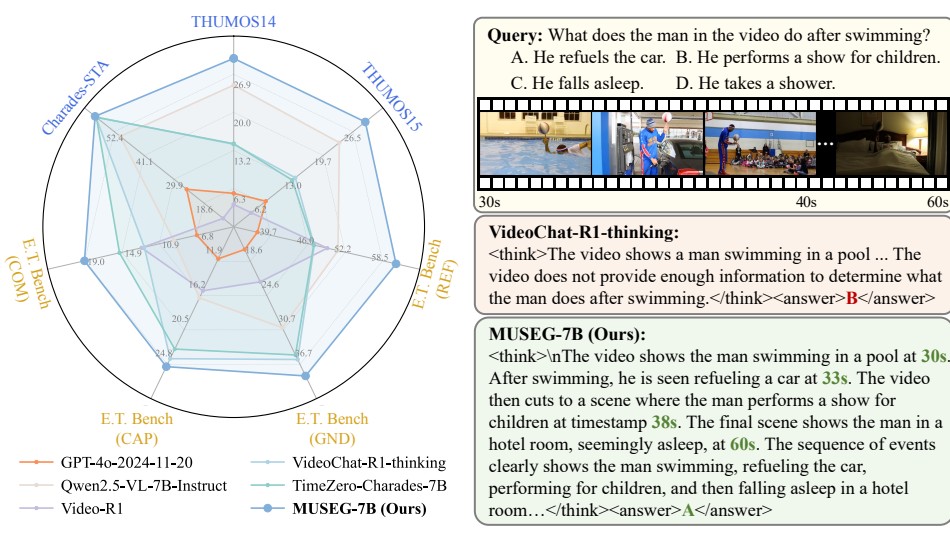

(a)                                                                                        (b)

Figure 1: (a) Performance of our MUSEG-7B on various temporal grounding (Charades-STA, THUMOS14 and THUMOS15) and broader time-sensitive video understanding (E.T. Bench Subset) tasks. (b) An example comparing our MUSEG-7B with previous models. MUSEG-7B performs more precise, timestamp-aware reasoning by effectively using multiple key temporal cues to derive the correct answer.

## 1 Introduction

Video temporal understanding (Liu et al., 2024a; Chen et al., 2024; Cheng et al., 2025b) refers to tasks of comprehending events based on temporal dynamics such as temporal grounding (Gao et al., 2017), dense video captioning (Wang et al., 2024), grounded video question answering (Xiao et al.,

2024), etc. This capability is vital for multimodal large language models (MLLMs) (Hurst et al., 2024; Team et al., 2023; Bai et al., 2025) in understanding complex temporal structures in videos and making accurate, context-aware predictions or decisions based on when and how events unfold.

Despite rapid progress and impressive results in general video understanding, current MLLMs still show significant limitations in temporal understanding (Liu et al., 2024b; Li et al., 2025c). Early efforts to address this are mainly based on supervised fine-tuning (SFT) to improve temporal comprehension (Bai et al., 2025; Liu et al., 2024a; Li et al., 2025a). As reinforcement learning (RL) has been shown to significantly improve complex reasoning and comprehension in large language models (LLMs) (Guo et al., 2025), recent studies have extended RL techniques to the video domain (Feng et al., 2025; Li et al., 2025b; Wang et al., 2025b; Zhang et al., 2025b), encouraging models to "reason before answering". This typically involves designing a format reward to ensure a structured reasoning process and an answer reward such as Intersection over Union (IoU) to measure the correctness of the predictions.

However, directly applying RL to video temporal understanding tasks has not achieved the same level of performance improvement as in textual domains (Feng et al., 2025; Li et al., 2025b). We attribute this limitation to two key challenges. First, most existing methods (Li et al., 2025b; Wang et al., 2025b) rely solely on single-segment temporal grounding, where each input query corresponds to only one video segment. This limits the ability to capture fine-grained, multi-segment temporal information, which is essential for complex video understanding tasks. Second, although temporal understanding depends fundamentally on reasoning over temporal cues, current RL approaches often fail to model them effectively. Reasoning process of current models typically consists of brief descriptions of video content, lacking detailed temporal analysis of key events, as illustrated in Figure 1 (b). Therefore, we argue that advancing MLLMs in video temporal understanding requires rethinking both the *training task design* and the *RL training recipe*.

In this paper, we propose timestamp-aware **MU**lti-**SE**gment **G**rounding (MUSEG), an RL-based method designed to enhance the temporal understanding and reasoning capabilities of MLLMs. On the task side, we incorporate *multi-segment grounding* into the training process, enabling models to learn from queries that align with multiple relevant video segments. This promotes stronger temporal understanding and better generalization to a wide range of time-sensitive tasks. On the training side, we introduce a customized RL training recipe with phased rewards, which progressively encourage the model to establish temporally grounded reasoning processes. Our recipe features a dedicated segment matching reward and a timestamp reward, encouraging models to perform fine-grained temporal reasoning over multiple segments as shown in Figure 1 (b). Additionally, we employ a multi-phase training strategy that balances guided learning and exploration. As illustrated in Figure 1 (a), MUSEG achieves significant improvements on temporal grounding benchmarks and generalizes effectively to other time-sensitive video understanding tasks. Our contributions can be summarized as follows:

- We propose MUSEG, a novel RL-based method for video temporal understanding, which enables MLLMs to reason over multiple temporally distributed events by incorporating multi-segment grounding into training.
- We design a tailored RL training recipe featuring novel reward functions and a multi-phase training strategy, effectively promoting fine-grained and temporally grounded reasoning.
- We conduct extensive experiments and analyses, showing that MUSEG consistently outperforms existing methods on video temporal understanding benchmarks, and validating the effectiveness of our task and training designs.

## 2 RELATED WORK

**Video Temporal Understanding.** Previous research on video temporal understanding mainly focuses on cross-references and alignments between videos and texts (Arnab et al., 2021; Luo et al., 2021; Liu et al., 2021; Xu et al., 2021; Wang et al., 2021). Recent advances in video temporal understanding have moved from these cross-modal attention-based local feature matching approaches to broader time-sensitive tasks, such as temporal grounding (Gao et al., 2017), dense video captioning (Wang et al., 2024), and grounded video question answering (Xiao et al., 2024). These methods attempt to fuse video temporal features and text features with LLMs to enhance model performance (Liu et al., 2024a; Li et al., 2025c; Yan et al., 2025).

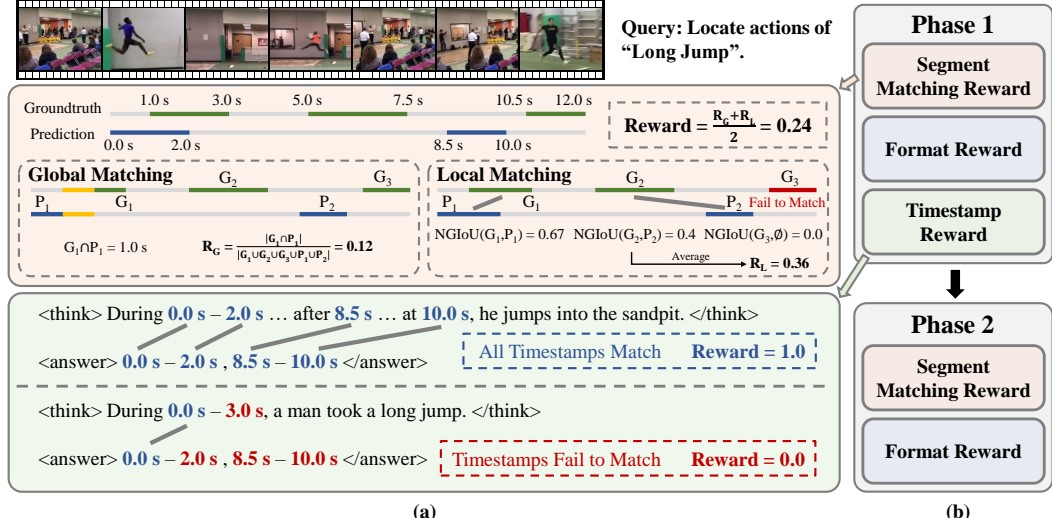

Figure 2: Overview of MUSEG. (a) Our proposed segment matching reward (up) and timestamp reward (down). (b) RL-based training process with phased rewards of MUSEG.

However, these methods involve automatic construction of Chain-of-Thoughts (CoTs) (Wei et al., 2022) in training data. Thus, they are limited by fixed reasoning patterns and potential issues with data quality, and model performance remains suboptimal on temporal understanding tasks, and struggle to generalize to complex scenarios (Liu et al., 2024a; Chen et al., 2024; Cai et al., 2024; Huang et al., 2024). Instead, we empower models to develop temporal-aware reasoning patterns autonomously, rather than constraining them to learn predefined patterns.

**RL for Video Understanding.** RL has been widely adopted in various textual tasks (Shao et al., 2024; Ouyang et al., 2022; Schulman et al., 2017). Recent works apply RL to general video question answering (Feng et al., 2025; Chen et al., 2025b; Dang et al., 2025; Wang et al., 2025a; Chen et al., 2025a; Zhang et al., 2025a) and temporal grounding tasks (Li et al., 2025b; Cheng et al., 2025a). However, these methods primarily provide rewards based on correctness of answers. The effective utilization of temporal information in the reasoning processes is not accounted for in the rewards, which may result in less effective CoTs. Models still struggle on complex temporal grounding tasks, and there is still room for improvement in generalizing to broader temporal understanding scenarios. In contrast, our approach imposes supervision on both reasoning processes and final answers.

## 3 PRELIMINARIES: REWARD DESIGN IN GRPO

Group Relative Policy Optimization (GRPO) (Shao et al., 2024) is a reinforcement learning (RL) training algorithm that has been widely adopted to enhance the reasoning abilities of large language models (LLMs). A key factor in GRPO is the design of the reward function. Prior work has demonstrated that verifiable rewards, though simple, can be highly effective in improving LLM reasoning (Guo et al., 2025; Yang et al., 2025). For instance, DeepSeek-R1 (Guo et al., 2025) incorporates two rule-based rewards:

- **Accuracy Rewards**: Assess whether the model produces correct outputs. For math tasks, correctness is verified by rule-based checkers, while for coding tasks, correctness is validated by compilers combined with pre-defined test cases.
- **Format Rewards**: Ensure that model outputs follow the required structure, namely the "`<think>`...`</think>` `<answer>`...`</answer>`" format.

While such rewards have proven effective in textual domains, we will show that directly applying them to video temporal understanding tasks leads to suboptimal performance.

## 4 METHOD

In this work, we propose a novel RL-based approach for video temporal understanding. The key components include multi-segment grounding as the training task (Section 4.1), a carefully designed

reward function with timestamp awareness (Section 4.2), and a new training paradigm with phased rewards (Section 4.3).

## 4.1 MULTI-SEGMENT GROUNDING TASK

Temporal grounding is the task that requires models to match text queries with corresponding video segments, which is capable of improving temporal understanding abilities of MLLMs (Liu et al., 2024a; Bai et al., 2025). Broadly, temporal grounding queries can be categorized into two types: *single-segment grounding*, where each query is associated with a single video segment, and *multi-segment grounding*, where a query may correspond to multiple video segments.

Previous RL-based temporal grounding approaches (Li et al., 2025b; Wang et al., 2025b) typically adopt single-segment grounding as the training task. However, our preliminary empirical study reveals that a notable portion of single-segment grounding questions can be solved through unintended shortcuts, for example, by identifying key objects rather than reasoning about the temporal structure of events. As shown in Table 1, this shortcut accounts for about 30% of cases, based on a manual check of 50 single-segment grounding questions sampled from E.T. Bench (Liu et al., 2024a). These findings suggest that relying solely on single-segment grounding tasks is insufficient for enhancing the temporal understanding abilities of MLLMs.

In contrast, multi-segment grounding queries are difficult to be answered by shortcuts, as shown in Table 1, so we add them to our training process. We ensure the number of single-segment grounding and multi-segment grounding queries are balanced, and our selected data are diverse in scenarios. Experiments demonstrate that incorporating multi-segment grounding significantly enhances model performance (see Section 5.3).

Table 1: Results of preliminary empirical study. We sample single-segment grounding and multi-segment grounding queries from E.T. Bench (Liu et al., 2024a), and examine whether they can be answered by shortcut of recognizing key objects.

| Query Type | w/ Shortcut | Total |
|---|---|---|
| Single-Segment | 15 | 50 |
| Multi-Segment | 4 | 50 |

## 4.2 REWARD DESIGN

We utilize segment matching reward (Section 4.2.1) to evaluate outputs for multi-segment grounding tasks, timestamp reward (Section 4.2.2) to stimulate model ability for temporal-aware reasoning, and format reward (Section 4.2.3) to encourage models to think before providing answers.

### 4.2.1 SEGMENT MATCHING REWARD

Segment matching reward is designed to align model outputs with ground truths. It consists of two parts, global matching and local matching, to enhance model abilities of understanding overall video contents, and grasping detailed events, respectively.

*Global matching* is shown in upper left area of Figure 2 (a). We measure the overlap ratio among all the ground truth segments $\{G_i\}$ and predicted segments $\{P_j\}$:

$$r_{\mathrm{G}} = \frac{\sum_{i,j} |G_i \cap P_j|}{|(\cup_i G_i) \cup (\cup_j P_j)|}, \text{ where } G_i \text{ and } P_j \text{ are represented as intervals.} \tag{1}$$

In the *local matching* process, we pair ground truths and predictions one-to-one as $\{(G_n, P_n)\}_{n=1}^N$, where $N = \max(|\{G_i\}|, |\{P_j\}|)$. As shown in upper right area of Figure 2 (a), we sort $\{G_i\}$ and $\{P_j\}$ according to their start timestamps, and match $G_k$ with $P_k$, where $k$ is the new index after sorting and $1 \leq k \leq \min(|\{G_i\}|, |\{P_j\}|)$. For the rest of ground truths or predictions, we match them with empty segments $\phi$. We also explore other matching strategies in Section 6.1. After matching, we leverage the normalized version of GIoU (Rezatofighi et al., 2019), denoted as NGIoU, as the metric to assess the overlap between paired ground truth $G_n$ and prediction $P_n$, where $1 \leq n \leq N$. The value of NGIoU ranges from 0 to 1 and is defined as follows:

$$\mathrm{NGIoU}(G_n, P_n) = \frac{1}{2} \left( 1 + \frac{|G_n \cap P_n|}{|G_n \cup P_n|} - \frac{|\mathcal{C} \backslash (G_n \cup P_n)|}{|\mathcal{C}|} \right), \tag{2}$$

where $\mathcal{C}$ is the shortest video segment covering $G_n$ and $P_n$. We use GIoU instead of IoU because it better guides model optimization when there is no overlap between the predicted video segment

and the ground truth. Specifically, to encourage the model outputs to align more closely with the ground truth, we impose a penalty when the number of predicted segments differs from that of the ground truth. This is implemented by setting the NGIoU score to 0 when paired with $\phi$, i.e., $\text{NGIoU}(\cdot, \phi) = 0$, and $\text{NGIoU}(\phi, \cdot) = 0$. Finally, the average NGIoU of all pairs is calculated:

$$r_{\text{L}} = \frac{\sum_{n=1}^{N} \text{NGIoU}(G_n, P_n)}{N} \tag{3}$$

And the final segment matching reward is defined as

$$r_{\text{M}} = \frac{r_{\text{G}} + r_{\text{L}}}{2} \tag{4}$$

### 4.2.2 TIMESTAMP REWARD

Explicitly incorporating temporal information into reasoning process during video comprehension helps models better understand complex temporal structures and events in videos, while neglecting temporal information may lead to misconceptions about video content (see the example in Figure 1 (b)). Previous works (Feng et al., 2025; Yu et al., 2025) reveal the importance of model ability of temporal-aware reasoning. Unfortunately, it remains a challenging problem to stimulate this ability.

To tackle this problem, we design the timestamp reward $r_{\text{T}}$ to enforce models to include timestamps which occur in the final answers in their reasoning processes. Suppose $\{T_{\text{A}}^i\}$ and $\{T_{\text{R}}^i\}$ are timestamps occurring in the answer and reasoning process of a model output, then

$$r_{\text{T}} = I_{\{T_{\text{R}}^i\} \subset \{T_{\text{A}}^i\}} \tag{5}$$

where $I$ is indicator function. As shown in lower part of Figure 2 (a), when all the timestamps occurring in the answer are found in thinking process, models get the reward. If some timestamps fail to match, the reward is set zero. With the timestamp reward, we encourage models to focus on temporal details during reasoning instead of thinking purely based on overall video contents.

### 4.2.3 FORMAT REWARD

Our format reward follows DeepSeek-R1 (Guo et al., 2025), enforcing models to output their thinking processes and final answers in format "`<think>...</think><answer>...</answer>`":

$$r_{\text{F}} = \begin{cases} 1, \text{if } o_i \text{ has right format} \\ 0, \text{otherwise} \end{cases} \tag{6}$$

### 4.3 TRAINING RECIPE WITH PHASED REWARDS

Our preliminary experiments indicate that explicitly encouraging models to output timestamps assists models in establishing a timestamp-aware reasoning strategy in early stages, but later leads to performance drop once the model becomes more capable. A possible reason is that enforcing such behavior restricts models to explore and develop more flexible reasoning strategies. To address this, we propose a training recipe with phased rewards, as illustrated in Figure 2(b).

In the early stage, all three rewards, segment matching reward $r_{\text{F}}$, timestamp reward $r_{\text{T}}$, and format reward $r_{\text{M}}$, are combined to form the final reward:

$$r_1 = [\beta r_{\text{T}} + (1 - \beta)r_{\text{F}}] + \alpha r_{\text{M}}. \tag{7}$$

In the later stage, we remove the timestamp reward to allow for more flexible reasoning patterns, yielding the following final reward:

$$r_2 = r_{\text{F}} + \alpha r_{\text{M}}. \tag{8}$$

Training with phased rewards leads to greater performance improvements than using either $r_1$ or $r_2$ alone throughout the entire process. Further analysis is provided in Section 6.2. Search of hyperparameters $\alpha$ and $\beta$ is introduced in Appendix C.

Table 2: Results of MLLMs on in-domain and out-of-domain tasks. *Results are copied from original paper. Detailed model versions and introduction of other baselines can be found in Appendix D.

| Model | In-Domain | | | | Out-of-Domain | | | | | | | | | |
| | Charades-STA (Single-Seg) | THUMOS14 (Multi-Seg) | THUMOS15 (Multi-Seg) | Perception Test (Multi-Seg) | E.T. Bench | | | | | E.T. Bench (Subset) | | | | |
| | | | | | REF | GND | CAP | COM | AVG | REF | GND | CAP | COM | AVG |
| **API-based Models** | | | | | | | | | | | | | | |
| GPT-4o | 25.1 | 5.5 | 6.7 | - | - | - | - | - | - | 37.4 | 16.5 | 11.6 | 6.8 | 18.1 |
| **Open-source ∼ 7B Models** | | | | | | | | | | | | | | |
| Qwen2.5-VL-7B | 50.2 | 24.9 | 23.4 | 25.3 | 53.1 | 30.7 | 16.2 | 11.3 | 27.8 | 51.0 | 30.3 | 16.5 | 9.3 | 26.8 |
| +vanilla SFT | 28.1 | 15.5 | 15.6 | 20.3 | 24.3 | 11.3 | 15.3 | 6.6 | 14.4 | 27.8 | 12.6 | 15.0 | 8.7 | 16.0 |
| +vanilla GRPO | 53.9 | 25.8 | 25.6 | 30.0 | 54.4 | 37.6 | 23.5 | 20.6 | 34.0 | 50.9 | 36.6 | 23.7 | 17.8 | 32.3 |
| E.T. Chat | 45.6 | 23.7 | 24.9 | 9.2 | 38.4* | **38.0*** | 16.7* | 13.5* | 26.7 | 31.8* | 33.8* | 17.1* | 11.1* | 23.5 |
| TRACE-7B | 29.9* | 7.6 | 7.8 | 14.0 | 33.6* | 33.8* | 20.3* | **25.8*** | 28.4 | 25.2 | 17.2 | 14.7 | 5.3 | 15.6 |
| Video-R1 | 11.3 | 3.5 | 3.4 | 5.7 | 50.3 | 25.3 | 15.6 | 12.4 | 25.9 | 49.2 | 22.2 | 15.6 | 12.8 | 25.0 |
| VideoChat-R1 | 59.4 | 14.3 | 13.4 | 27.1 | 55.8 | 35.6 | 22.1 | 19.5 | 33.3 | 47.0 | 35.9 | 24.1 | 12.5 | 29.9 |
| TimeZero | 59.2 | 14.4 | 12.7 | 26.8 | 55.9 | 35.8 | 21.4 | 17.1 | 32.6 | 46.9 | 35.1 | 22.9 | 15.2 | 30.0 |
| MUSEG-7B (Ours) | **59.7** | **29.7** | **29.3** | **31.7** | **61.9** | 37.5 | 23.7 | 24.0 | **36.8** | **60.8** | **38.8** | **25.1** | **19.0** | **35.9** |
| **Open-source ∼ 3B Models** | | | | | | | | | | | | | | |
| Qwen2.5-VL-3B | 41.4 | 12.6 | 12.8 | 19.4 | 51.7 | 20.4 | 13.6 | 8.0 | 23.4 | 52.9 | 20.4 | 12.7 | 7.6 | 23.4 |
| TEMPURA | 44.5 | 8.7 | 12.1 | 20.7 | 46.3 | 26.1 | 14.4 | **10.2** | 24.3 | **56.4** | 22.8 | 13.3 | 3.5 | 24.0 |
| MUSEG-3B (Ours) | **53.7** | **21.0** | **20.3** | **29.1** | **53.9** | **30.0** | **18.7** | 8.8 | **27.9** | 54.3 | **28.7** | **18.3** | **11.8** | **28.3** |

# 5 EXPERIMENTS

## 5.1 IMPLEMENTATIONS

Our training dataset is constructed from E.T. Instruct 164k (Liu et al., 2024a) and Charades-STA (Gao et al., 2017). For E.T. Instruct 164k, we only sample data from temporal video grounding (TVG) and temporal action localization (TAL) tasks. Our final training dataset consists of 12.6k samples, including 6,967 with a single segment and 5,633 with multiple segments as ground truths.

We train MUSEG-7B and MUSEG-3B based on Qwen2.5-VL-7B-Instruct and Qwen2.5-VL-3B-Instruct (Bai et al., 2025) (abbreviated as Qwen2.5-VL-7B and Qwen2.5-VL-3B in Table 2), respectively. They are trained with timestamp reward ($r_1$) for 400 steps and without timestamp reward ($r_2$) for another 500 steps. For comparison, we also conduct SFT and naive RL experiments on Qwen2.5-VL-7B-Instruct with our constructed dataset as baselines. For naive RL experiment, we remove timestamp reward, and retain only global matching part of segment matching reward as well as format reward. All other settings remain consistent with those used for training MUSEG-7B. Training details can be found in Appendix B.

## 5.2 BENCHMARKS AND EVALUATION METRICS

We evaluate MUSEG-7B and MUSEG-3B on both temporal grounding tasks (in-domain) and broader time-related tasks (out-of-domain). For in-domain evaluation, we use the Charades-STA (Gao et al., 2017) test set for single-segment grounding and report performance using mIoU. For multi-segment grounding, we adopt the validation sets of THUMOS14, THUMOS15 (Idrees et al., 2017), and Perception Test (Patraucean et al., 2023), and measure F1 scores averaged over four IoU thresholds (0.1, 0.3, 0.5, and 0.7), following previous work (Liu et al., 2024a).

For out-of-domain evaluation, we assess model generalization on a variety of time-related tasks in E.T. Bench (Liu et al., 2024a), including referring (REF), grounding (GND), dense captioning (CAP), and complex understanding (COM). We adopt the metrics from the original benchmark: accuracy for referring, F1 score for grounding, sentence similarity for dense captioning, and recall for complex understanding.

## 5.3 MAIN RESULTS

As shown in Table 2, MUSEG-7B and MUSEG-3B outperform other SFT- or RL-based methods on most in-domain and out-of-domain tasks among all ∼ 7B and ∼ 3B models, and even surpass GPT-4o. Our method shows a significant advantage over base models. MUSEG-7B achieves more than 10% performance enhancement on all the tasks compared to its base model Qwen2.5-VL-7B-Instruct. Also, it is worth noting that our model gets doubled performance on complex understanding task, showing strong ability of generalization.

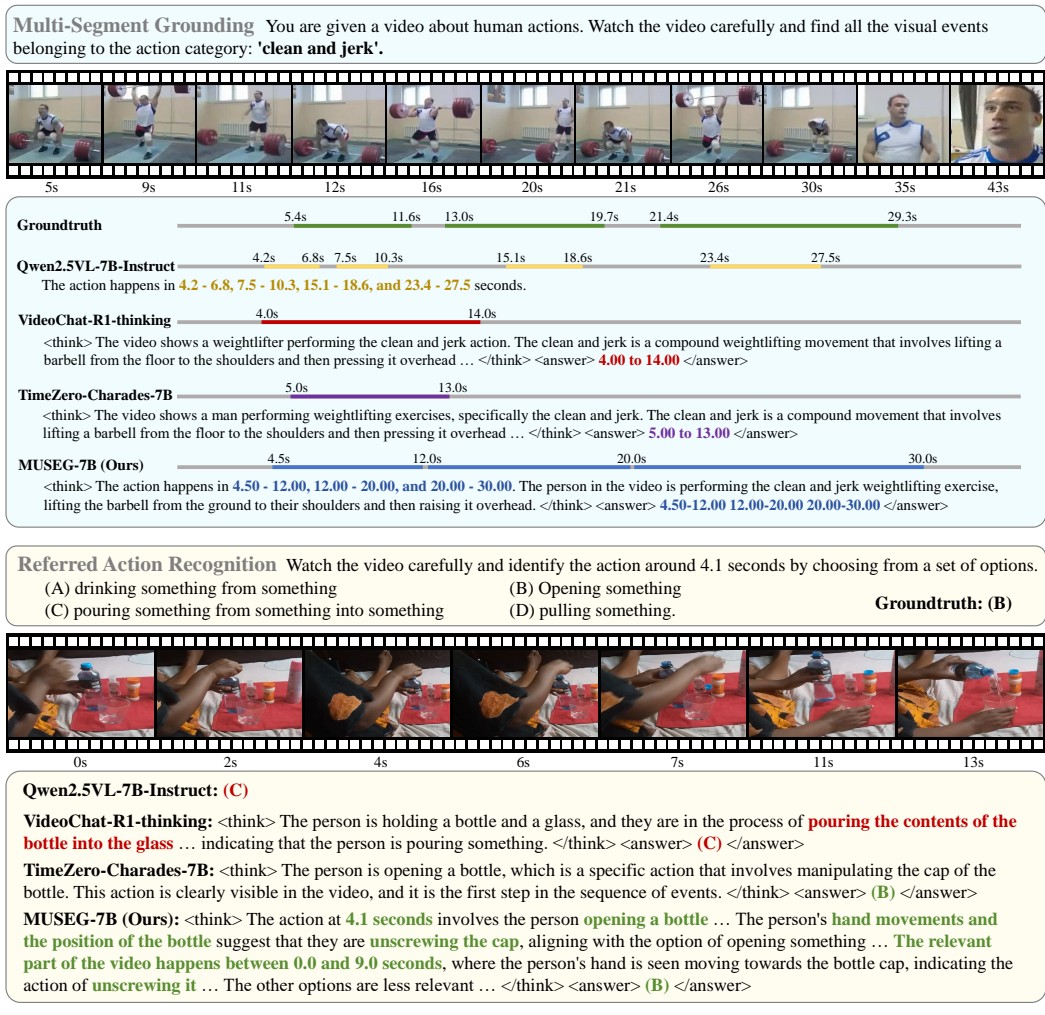

Figure 3: Cases of our MUSEG-7B and baselines on multi-segment grounding (in-domain) and referred action recognition (out-of-domain) tasks.

Furthermore, we compare MUSEG-7B with vanilla GRPO baseline ("+vanilla GRPO" in Table 2), which is trained on the same data as MUSEG-7B, and with VideoChat-R1 ("VideoChat-R1" in Table 2), which is trained exclusively on single-segment grounding using vanilla GRPO with Charades-STA as the training dataset. As depicted in Table 2, VideoChat-R1 experiences a performance decline on multi-segment grounding tasks, whereas the vanilla GRPO baseline with multi-segment grounding task exhibits limited performance enhancement. This suggests that simply introducing multi-segment grounding during training is insufficient to yield substantial gains. In contrast, our MUSEG-7B delivers significantly larger gains across both grounding tasks and a broader set of time-sensitive tasks, while maintaining comparable general video QA performance to its base model (see Appendix E). These results further demonstrate that the effectiveness of MUSEG-7B stems from its innovative design of training tasks and reward recipes.

Qualitative case studies presented in Figure 3 provide additional evidence of the effectiveness of our proposed model. The first case is a multi-segment grounding task (in-domain) with the query "clean and jerk". VideoChat-R1-thinking and TimeZero-Charades-7B only recognize the video segment corresponding to the first attempt, consistent with the fact that they are trained only with single-segment grounding tasks. In contrast, MUSEG-7B accurately localizes all three weight-lifting attempts. The performance gap highlights effectiveness of multi-segment grounding training tasks.

The second case involves referred action recognition (out-of-domain) query about event happening around 4.1 seconds. Seen from the video, the person first opens the bottle, and then pour water out from it. VideoChat-R1 incorrectly aligns the event of pouring water from the bottle (occurring at 11 seconds) with a 4.1-second timestamp, demonstrating a temporal misalignment in its reasoning.

Table 3: Results with different matching strategies. For all the experiments, we train Qwen2.5-VL-7B for 900 steps same as the training process of MUSEG-7B.

| Local Matching Strategy | Charades-STA | THUMOS14 | THUMOS15 | E.T. Bench (Subset) | | | | |
|---|---|---|---|---|---|---|---|---|
| | | | | REF | GND | CAP | COM | AVG |
| w/o Local Matching | 59.3 | 26.0 | 25.7 | 46.2 | 37.0 | 22.5 | 15.3 | 30.3 |
| w/ Local Matching (Maximum) | 58.2 | 28.6 | 28.2 | 56.2 | 31.5 | 24.7 | 17.4 | 32.5 |
| w/ Local Matching (Sequential) | **59.7** | **29.7** | **29.3** | **60.8** | **38.8** | **25.1** | **19.0** | **35.9** |

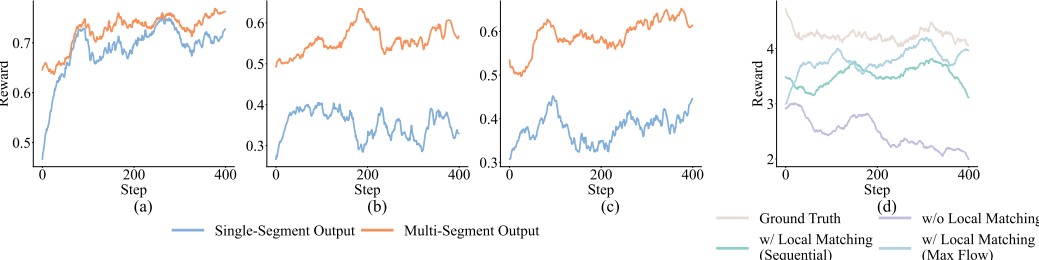

Figure 4: Segment matching reward (a) w/o local matching, (b) w/ local matching (sequential), and (c) w/ local matching (maximum). (d) Evolution of numbers of predicted segments during training process. For all the plots, we only consider queries whose ground truths are more than one segments.

TimeZero-Charades-7B provides the correct answer but lacks precise timestamp references in its explanation. In contrast, MUSEG-7B exhibits superior temporal reasoning capability: it not only identifies the bottle-opening action around 4.1 seconds but also accurately localizes the corresponding video segment.

## 6 ANALYSES

### 6.1 LOCAL MATCHING STRATEGIES

We delve deeper to verify effectiveness of local matching in segment matching reward. We conduct experiments of removing local matching, and only keeping global matching ("w/o Local Matching" in Table 3). Additionally, we explore another design, which involves matching ground truths and predictions to maximize average overlap ("w/ Local Matching (Maximum)" in Table 3). We do this by calculating maximum weighted matching in bipartite graph. For ground truth segments $\{G_i\}$ and predicted segments $\{P_j\}$, we construct a complete bipartite graph $\mathcal{G}$ as

$$\mathcal{G} = \{(G_i, P_j, W_{i,j})\}, \tag{9}$$

where $(G_i, P_j, W_j)$ denotes an edge connecting $G_i$ and $P_j$ with weight $W_{i,j} = \text{NGIoU}(G_i, P_j)$, then we calculate $r_L$ as follows:

$$r_{\text{L}} = \frac{\text{Matching}(\mathcal{G})}{\max(|\{G_i\}|, |\{P_j\}|)} \tag{10}$$

where $\text{Matching}(\cdot)$ is the maximum weighted matching function. Table 3 shows that including local matching boost overall model performance compared to only keeping global matching. Additionally, sequential matching reaches better performance than maximum matching. Therefore, we finally adopt sequential matching in MUSEG.

We also notice that drops of model performance on multi-segment grounding are much larger than single-segment grounding when local matching is removed. To better understand its reason, we examine differences in rewards model would get when it produces a single segment or at least two segments for a query whose ground truth consists of more than one segments. As shown in Figure 4 (a), (b), and (c), local matching strategies impose significant penalties on segment matching rewards when model output only contains a single segment, but the penalties imposed by global matching are relatively weak. We further report evolution of numbers of predicted segments during training process in Figure 4 (d). When we remove local matching, numbers of predicted segments significantly drop and their gaps from ground truths become larger. This indicates that local matching can help better align numbers of predicted segments with ground truths.

Table 4: Results with different training recipes.

| Training Paradigms | Charades-STA | THUMOS14 | THUMOS15 | E.T. Bench (Subset) | | | | |
|---|---|---|---|---|---|---|---|---|
| | | | | REF | GND | CAP | COM | AVG |
| w/o Timestamp Reward | 56.9 | 28.4 | 28.3 | 55.1 | 37.6 | 22.3 | 13.2 | 32.1 |
| w/ Timestamp Reward | 57.3 | 26.1 | 24.6 | 57.3 | 28.9 | 22.0 | 16.1 | 31.1 |
| w/ Timestamp Reward for 400 Steps | **59.7** | **29.7** | **29.3** | **60.8** | **38.8** | **25.1** | **19.0** | **35.9** |

## 6.2 DESIGN OF PHASED REWARDS

In this section, we explore the effectiveness of our proposed training recipe with phased rewards. We compare it against training model with or without timestamp reward during the whole training process in Table 4. From the table we can see that our proposed recipe of training the model with timestamp reward for 400 steps and without timestamp reward for another 500 steps reaches the highest performance. We further change the total training steps and report the results in Figure 5 (a). We can see that our proposed recipe consistently outperforms other training strategies, showing effectiveness over different data scales. We also explore model performance when we vary number of steps of keeping timestamp reward. Figure 5 (b) shows that when the model is trained with timestamp reward for 400 steps, its performance reaches the peak.

To understand the underlying reasons, we examine values of segment matching reward, which reflects accuracy of model output, throughout the training process. As illustrated in Figure 6, when the model is trained either without the timestamp reward or with the timestamp reward applied throughout the entire training process, there is minimal improvement in performance after 400 steps as reasoning forms of models stabilize. In contrast, if the timestamp reward, which is designed to guide the model in referencing specific timestamps during reasoning, is removed after 400 steps, the model can continue to freely explore more effective reasoning strategies, leading to continuous enhancement in its segment matching reward in subsequent steps.

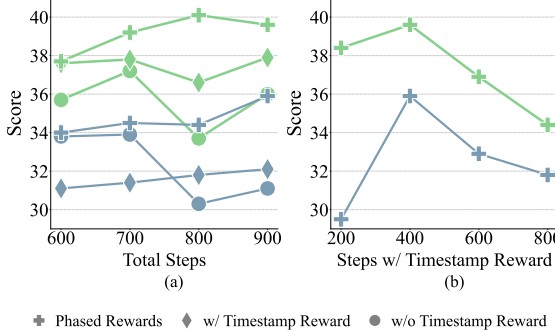

Figure 5: (a) Model performance with different training recipes. For setting of phased rewards, we train models with timestamp reward for 300 steps when total steps are 600 and 700, for 400 steps when total steps are 800 and 900. (b) Model performance when we vary number of steps with timestamp reward, keeping total steps to be 900. For all the experiments, we report average score of Charades-STA, THUMOS14 and THUMOS15 as in-domain score, and average score of E.T. Bench (Subset) as out-of-domain score.

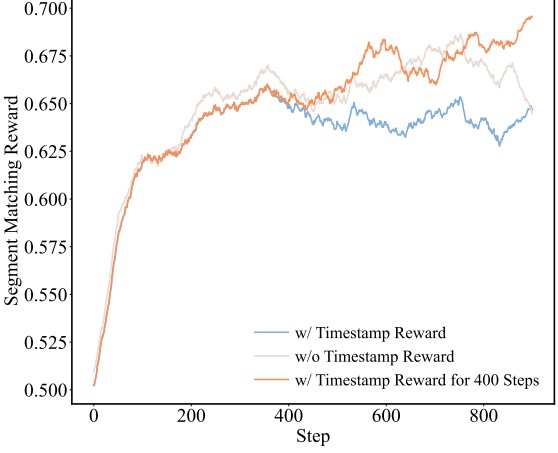

Figure 6: Rewards with different training recipes. We also report timestamp reward during training.

## 7 CONCLUSION

In this work, we introduce MUSEG, a RL-based method to improve video temporal understanding abilities of MLLMs. Experiments demonstrate effectiveness of our method on improving model performance on single-segment and multi-segment grounding tasks, as well as broader time-sensitive scenarios. We hope our proposed method will inspire future research on enhancing temporal understanding abilities of MLLMs.

## REPRODUCIBILITY STATEMENT

We train MUSEG-7B and MUSEG-3B using publicly available models and datasets. Implementation details can be found in Section 5.1 and Appendix B. Additionally, we include our training and inference code in supplementary material to enhance reproducibility.

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

## A  LLM Usage Statement

Throughout the completion of this work, LLMs are solely used for the purpose of spelling checking and grammatical error detection for the manuscript. They are not employed for any other purposes, such as deriving idea of this paper, or validating the methods or results.

## B  Implementation Details

We leverage 7B and 3B models of Qwen2.5-VL (Bai et al., 2025) series as our base models. They are trained on large scale image and video data and demonstrate strong instruction following and reasoning abilities. Additionally, there are special designs in Qwen2.5-VL to enable models to process absolute timestamps and dynamic resolutions of video frames. During training and inference of MUSEG-7B and MUSEG-3B, we set maximum total video tokens to be 3584 and maximum number of frames to be 448.

We train MUSEG-7B and MUSEG-3B for 900 steps in total, including 400 steps with timestamp reward and another 500 steps without timestamp reward. We set $batch\_size = 14$ and $learning\_rate = 1e-5$. We set $\alpha = 2$ in phase 1 and phase 2 reward, and $\beta = 0.4$ in phase 1 reward (see experiments of hyperparameters search in Appendix C). Considering that base models have been trained on temporal-related data and already have strong abilities of instruction-following, we do not include SFT stage in our experiments as DeepSeek-R1 (Guo et al., 2025).

It takes about 22 hours for MUSEG-7B phase 1 training, 27 hours for MUSEG-7B phase 2 training, 16 hours for MUSEG-3B phase 1 training and 20 hours for MUSEG-3B phase 2 training on 8 A100-80G GPUs.

## C  Hyperparameters Search

For $\alpha$ and $\beta$, we conduct experiments using a small dataset comprising 1400 samples for training, along with a randomly selected subset of 200 samples from THUMOS14 and THUMOS15 for evaluation, aiming at identifying the optimal combination of hyperparameters. Based on results presented in Table 5 and Table 6, we select the best combination of $\alpha = 2$ and $\beta = 0.4$.

Table 5: Model performance with different settings of $\alpha$.

| Hyperparameters | THUMOS (subset) |
|---|---|
| $\alpha = 1$, $\beta = 0.4$ | 23.4 |
| $\alpha = 1.5$, $\beta = 0.4$ | 27.6 |
| $\alpha = 2$, $\beta = 0.4$ | **28.8** |
| $\alpha = 3$, $\beta = 0.4$ | 19.3 |
| $\alpha = 4$, $\beta = 0.4$ | 21.5 |

Table 6: Model performance with different settings of $\beta$.

| Hyperparameters | THUMOS (subset) |
|---|---|
| $\alpha = 2$, $\beta = 0.2$ | 28.4 |
| $\alpha = 2$, $\beta = 0.4$ | **28.8** |
| $\alpha = 2$, $\beta = 0.6$ | 28.7 |

## D  Introduction of Baselines

Our baselines can be categorized into SFT-based methods and RL-based methods. We introduce SFT-based models first:

**E.T. Chat ($\sim$ 7B)**: It compresses video frames into single tokens using a Q-Former-based compressor with cross-attention, and generates timestamps with special tokens. It is trained on E.T. Instruct 164k, a dataset covering 9 tasks across 14 sources.

**TRACE ( ∼ 7B)**: It is trained with a causal event modeling framework, integrating timestamp, salient score, and textual caption prediction tasks. Its training data include 1.9M samples from Valley, TextVR, ShareGPT4Video, and 0.9M samples form ActivityNet Captions and InternVid.

**TEMPURA ( ∼ 3B)**: It is trained with masked event prediction reasoning, event segmentation and dense captioning tasks. Its training data consist of 500k samples.

Then we introduce RL-based models:

**Video-R1 ( ∼ 7B)**: It is trained by SFT with 165k samples and RL with 260k samples. Its training data consist of various general image question answering and video question answering tasks.

**VideoChat-R1-thinking ( ∼ 7B, abbreviated as VideoChat-R1 in Table 2)**: It is trained with temporal grounding, object tracking, video captioning and grounded video question answering tasks, with a total data scale of 18.0k samples.

**TimeZero-Charades-7B ( ∼ 7B, abbreviated as TimeZero in Table 2)**: It is trained towards temporal grounding tasks. A version of its models is trained with Charades-STA (Gao et al., 2017).

Additionally, we report performance of GPT-4o-2024-11-20 (Hurst et al., 2024) (abbreviated as GPT-4o in Table 2) for reference. In consideration of inference costs, we do not report results of GPT-4o on Perception Test and the whole set of E.T. Bench. Only results on a subset of 470 samples of E.T. Bench, specified by the original paper, are reported.

# E  MODEL PERFORMANCE ON GENERAL VIDEO QA TASKS

We present model performance on general video QA benchmarks, MVBench (Li et al., 2024) and Video-MME (Fu et al., 2025), in Table 7. Results indicate that our overall performance on MVBench and Video-MME is comparable to that of our base model. Our approach does not negatively impact model performance on general video QA tasks.

Table 7: Model performance on MVBench and Video-MME.

| Model | MVBench | Video-MME | | | |
| --- | --- | --- | --- | --- | --- |
| | | SHORT | MEDIUM | LONG | AVG |
| Qwen2.5-VL-7B-Instruct | 65.7 | 71.8 | 62.7 | 52.6 | 62.4 |
| MUSEG-7B | **67.4** | **71.9** | **63.7** | **53.6** | **63.1** |

# F  TRAINING PROMPTS

We prompt models to include timestamps in their reasoning processes and ensure that the timestamps are consistent with those in answers using instruction in Table 8.

Table 8: Training prompts.

**Training prompts**

{QUESTION} First, output reasoning process in `<think> </think>` tags. The reasoning process must REFER TO SPECIFIC TIMESTAMPS TO TELL WHERE YOU GET THE INFORMATION FROM THE VIDEO. Then summarize your reasoning process above and output selected segments like `<answer>X.XX-X.XX</answer>`, where X denotes arabic numbers. If there are multiple segments, separate them with spaces like `<answer>X.XX-X.XX X.XX-X.XX</answer>`. Your output format should be like `<think>...</think><answer>...</answer>`.

