# OpenReview forum: "MUSEG: Reinforcing Video Temporal Understanding via Timestamp-Aware Multi-Segment Grounding"
_ICLR.cc/2026/Conference — Submitted to ICLR 2026_

### Official Review · Reviewer_5XGo · 2025-10-15

**Soundness:** 3
**Presentation:** 3
**Contribution:** 2
**Rating:** 4
**Confidence:** 3

**Summary:**

This paper introduces MUSEG, a reinforcement learning (RL) method to improve the fine-grained temporal understanding of Multimodal Large Language Models (MLLMs) in videos.

The core idea is timestamp-aware multi-segment grounding, which enables the model to align a query with multiple relevant video segments. This is optimized through a customized RL training recipe using phased rewards that progressively guides the model's reasoning.

Experiments show MUSEG significantly outperforms existing methods on temporal grounding and time-sensitive video QA tasks, indicating strong performance and generalization.

**Strengths:**

With clear motivation and intuition, the paper uses GRPO and introduces two new reward functions, allowing the model to learn how to correctly segment events within videos for better temporal understanding. Especially in the segment matching reward, the authors meticulously design a function that computes the IoU of the predicted timestamps compared to the ground truth, a sound and wise choice.   Finetuning models with this method demonstrates improvements on selected temporal benchmarks. The authors also conducted exhaustive ablations in an attempt to justify different training strategies and reward design.

**Weaknesses:**

1. Though the authors presented promising results on 4 different benchmarks, it would be great if they could evaluate on more temporal benchmarks, such as TempCompass [1], TemporalBench, Vinoground [2]'s video score (which contains multiple segments), and so on, if time permits.
2. In Section 4.2.2, the timestamp reward is designed to be 1 if all timestamps are correct, and 0 if any are wrong. This design is concerning to me because I do not understand why this reward is so strict; for segment matching the authors applied a very lenient strategy involving IoU, why won't one consider giving partial credit to the timestamp reward as well? I believe this might be the reason why the authors had to remove timestamp reward after 400 steps.
3. In Section 6.2, the authors stated that "the model can continue to freely explore more effective reasoning strategies" by removing the timestamp reward mid-training. Though the improvements quantitatively are acknowledged, there is no qualitative result demonstrating the difference in reasoning between removing and not removing the reward.
4. Accompanied by the above doubts about the reward function, which is the core contribution of this paper, I hate to say that I don't think the paper has enough novelty. But this idea is subject to change depending on the rebuttal for the above points.

[1] Liu et al, 2024, TempCompass: Do Video LLMs Really Understand Videos?

[2] Zhang et al, 2024, Vinoground: Scrutinizing LMMs over Dense Temporal Reasoning with Short Videos

**Questions:**

See weaknesses.

---

> ### Author Response · Authors · 2025-11-23
> **Responses**
>
> We sincerely appreciate your thoughtful comments and concerns. Our responses are as follows.
>
> **Weakness 1: Evaluation Across More Datasets**
>
> Please refer to Common Concern 1. Thank you.
>
> **Weakness 2: Experiment of Giving Partial Credit to the Timestamp Reward**
>
> Thank you for the suggestion. We report the results of allowing partial credit for the timestamp reward throughout the entire training process, instead of using our phased reward recipe. However, based on our observations, this variant performs worse than MUSEG-7B, and the model frequently fails to correctly predict multiple segments when the ground truth contains more than one.
>
> | Model | Charades-STA | THUMOS14 | THUMOS15 | E.T. Bench (REF) | E.T. Bench (GND) | E.T. Bench (CAP) | E.T. Bench (COM) | E.T. Bench (AVG) |
> |---|:---:|:---:|:---:|:---:|:---:|:---:|:---:|:---:|
> | Qwen2.5-VL | 50.2 | 24.9 | 23.4 | 51.0 | 30.3 | 16.5 | 9.3 | 26.8 |
> | VideoChat-R1-Thinking | 59.4 | 14.3 | 13.4 | 47.0 | 35.9 | 24.1 | 12.5 | 29.9 |
> | MUSEG-7B-Partial | 56.7 | 19.7 | 20.0 | 53.4 | 33.1 | 22.8 | 18.2 | 31.9 |
> | MUSEG-7B | **59.7** | **29.7** | **29.3** | **60.8** | **38.8** | **25.1** | **19.0** | **35.9** |
>
> * We utilize the official subset of 470 samples from E.T. Bench for our evaluation.
>
> **Weakness 3: Qualitative Results**
>
> We present a qualitative example below from E.T. Bench to illustrate this effect. Removing timestamp reward during the middle stage of training allows the model to engage in timestamp-based reasoning more flexibly, ultimately leading to better predictions.
>
> Specifically, in this case, MUSEG-7B is no longer restricted to focusing only on actions around the 9.0-second mark; instead, it can examine a broader and more relevant segment of the video to better capture temporal information.
>
> ```
> Question: You are given a video containing a series of actions. Watch the video carefully and identify the action around 9.0 seconds by choosing from a set of options.
>     (A) covering something with something    (B) opening something
>     (C) writing or drawing something on a piece of paper    (D) showing something to the camera.
>
> Ground Truth: (C)
>
> Phased Rewards
> <think>The action at 9.0 seconds is writing or drawing something on a piece of paper. The person is holding a marker and writing on a piece of paper, which is a clear indication of writing or drawing. The relevant part of the video happens in 4.0 - 14.0 seconds.</think><answer>C</answer>
>
> w/ Timestamp Reward Throughout Training
> <think>At 9.0 seconds, the person is holding a piece of paper with the word "REF" written on it. This suggests that the person is showing something to the camera, as they are holding the paper up for the viewer to see. The action of showing the paper is the most relevant to the given options. The other options do not fit the context of the video at this timestamp.</think><answer>D</answer>
> ```

---

> > ### Comment · Reviewer_5XGo · 2025-11-23
> >
> > I appreciate the authors' response and I think my questions 1 and 3 have been resolved. I want to continue asking about question 2: what do you think is the possible cause behind the performance drop using partial credit? What exactly is the formula you use for partial credit?

---

> > > ### Author Response · Authors · 2025-11-25
> > > **Responses**
> > >
> > > Thank you for the follow-up questions. We compute the ratio of timestamps in the final answer that also appear in the reasoning process. Let $T_A$ and $T_R$ denote the timestamps in the answer and in the reasoning process, respectively. We define $$r_T=\frac{|T_R\cap T_A|}{|T_A|},$$ and the remaining rewards follow those used in MUSEG-7B.
> > >
> > > A possible explanation is that the partial-credit signal is not strong enough to encourage the model to build a temporally grounded reasoning chain. We observe that in a number of cases, the reasoning process contains no timestamps, and in other cases, some timestamps in the final answer do not appear in the reasoning process, leaving those segments without supporting rationales and leading to errors.

---

> > > > ### Comment · Reviewer_5XGo · 2025-11-25
> > > >
> > > > Very interesting discovery! I have no further questions and will raise my score to 6.

---

> > > > > ### Author Response · Authors · 2025-11-26
> > > > > **Responses**
> > > > >
> > > > > Many thanks for your insightful suggestions! We will revise these points accordingly in our next version.

---

### Official Review · Reviewer_d1dA · 2025-10-26

**Soundness:** 4
**Presentation:** 3
**Contribution:** 3
**Rating:** 4
**Confidence:** 3

**Summary:**

This paper introduces a RL framework called MUSEG. The proposed method improves MLLMs' ability to understand temporal events in videos. Specifically, this paper proposed the task "multi-segment grounding", and uses GRPO with "segment matching reward" and "timestamp reward" to align predicted with true video segments and encourage temporal reasoning. Experiments show that MUSEG significantly outperforms previous SFT- and RL-based models on multiple benchmarks, achieving stronger fine-grained temporal reasoning and better generalization to time-sensitive video tasks.

**Strengths:**

1. This paper's proposed methodology has a very clear motivation. It performs a preliminary empirical study revealing why using single-segment grounding as the training task won't work.
2. The proposed approach outperforms previous baseline models on multiple benchmarks with both 3B and 7B models.
3. This paper provides comprehensive analysis into why the proposed approach would work, along with the strength (1), making the proposed approach more reasonable.
4. The proposed method won't hurt the model's performance on general video QA task.

**Weaknesses:**

1. The proposed method's performance improvement on general task is greatly limited (compared with the performance improvement on grounding task). On all benchmarks, the performance improvement is less than 2%. The result reveals that there's a great limitation on the method's generalizability.
2. The idea of utilizing the multi-segment as task has already appeared in the related work [1], which appears on the Internet on Jun 23 2025. It is one month earlier than the ICLR's comparison cutoff July 24, 2025, therefore, I think the connection to this work needs to be discussed, and it's likely that this work[1] needs to be compared with the proposed approach.

[1] Universal Video Temporal Grounding with Generative Multi-modal Large Language Models, NeurIPS 2025

**Questions:**

1. In Table 2, is the "+vanilla GRPO" based on "+vanilla SFT" (so it's + GRPO AND +SFT), or is based directly on the original "Qwen2.5-VL-7B"?
2. Why "seconds" is chosen as unit in verbal reasoning, rather than frames, millioseconds or minutes. If this will cause issues for long video or extremely short video?

---

> ### Author Response · Authors · 2025-11-23
> **Responses**
>
> We sincerely appreciate your thoughtful comments and concerns. Our responses are provided below.
>
> **Weakness 1: Generalizability of Our Method**
>
> Please refer to Common Concern 2. Thank you.
>
> **Weakness 2: Novelty of Utilizing Multi-segment Tasks**
>
> We do not claim to be the first to utilize multi-segment grounding tasks. Our contribution lies in analyzing the importance of incorporating multi-segment grounding within RL training, and in designing novel reward functions and a phased training recipe that enable models to reason more effectively over temporal cues. We will include the relevant prior work [1] in a future revision.
>
> **Question 1: Setting of Vanilla GRPO**
>
> We apply vanilla GRPO directly to Qwen2.5-VL-7B-Instruct without SFT, consistent with the setting used for MUSEG-7B. Since the base models have already been trained on temporally related data and possess strong instruction-following abilities, we do not include an SFT stage in our experiments, as stated in Appendix B.
>
> **Question 2: Handling Videos of Varying Lengths**
>
> Please refer to Common Concern 3. Thank you.

---

> > ### Comment · Reviewer_d1dA · 2025-11-23
> >
> > Thank you very much for your response. However, I still have concerns about the comparison between MUSEG and UniTime [1]. Although your method adopts a different approach from [1], both works address the same task of video temporal grounding. Therefore, it seems important to provide a direct performance comparison between the two methods and to discuss their relative applicability - e.g., in what scenarios one would prefer MUSEG over UniTime.
> >
> > Moreover, the reported performance of MUSEG appears to be substantially lower than that of UniTime on the same datasets (Charades-STA, ActivityNet-Captions), which weakens the current SOTA claim. Could you more clearly distinguish the contributions of your work beyond UniTime and elaborate on what is novel or advantageous about MUSEG in light of this performance gap?
> >
> > [1] Universal Video Temporal Grounding with Generative Multi-modal Large Language Models, NeurIPS 2025

---

> > > ### Author Response · Authors · 2025-11-25
> > > **Responses**
> > >
> > > Thank you for the follow-up questions. We compare the performance of MUSEG and UniTime in the table below. For single-segment grounding, MUSEG outperforms UniTime-Zero but remains below UniTime-Full. A likely reason is that the three models are trained on progressively larger amounts of same-origin data—UniTime-Zero, MUSEG-7B, and UniTime-Full—with exact scales shown in square brackets, and their performance rises in the same order. For multi-segment grounding, MUSEG-7B achieves a substantial performance gain while none of the models are trained on source-identical data, highlighting the strength of our model design and its superior ability to capture multiple visual cues in multi-segment grounding tasks.
> > >
> > > | Model  | Charades-STA (Single, R1@.5 / R1@.7 [data]) | ANet-Captions (Single, R1@.5 / R1@.7 [data]) | THUMOS14 (Multi, Mean F1) | THUMOS15 (Multi, Mean F1) |
> > > |:---|:---:|:---:|:---:|:---:|
> > > | UniTime-Full | 75.27 / 56.85 [11.2 K] | 53.67 / 35.90 [37.4 K] | 13.64 | 12.96 |
> > > | UniTime-Zero | 59.09 / 31.88 [0.0 K] | 22.77 / 14.14 [0.0 K] | / | / |
> > > | MUSEG-7B | 70.54 / 47.63 [6.3 K] | 42.99 / 23.71 [0.8 K] | 29.67 | 29.35 |
> > >
> > > * As UniTime-Zero is not publicly released, we cannot report its performance on THUMOS14 and THUMOS15.
> > >
> > > * For Charades-STA, our paper reports mIoU. Here, we additionally compute R1@.5 and R1@.7 to enable a fair comparison with UniTime.
> > >
> > > * For ANet-Captions, the evaluation protocol and data usage differ between our results in Common Concern and those reported by UniTime. We follow the original ANet-Captions format by treating it as a multi-segment grounding task, providing multiple queries and segments per sample, and report mean F1 scores. In contrast, UniTime decomposes each sample into single-query, single-segment cases and reports R1@.5 and R1@.7, treating it as a single-segment task. Moreover, we report results on the val 1 (test) split, while UniTime reports on val 2. To ensure comparability, the table above aligns our settings with those used in UniTime.

---

> > > > ### Comment · Reviewer_d1dA · 2025-11-25
> > > >
> > > > Thank you very much! The results look good, so I increased my ratings to 6.

---

> > > > > ### Author Response · Authors · 2025-11-25
> > > > > **Responses**
> > > > >
> > > > > We are glad our responses addressed your concerns! We will update the discussions in future version.

---

### Official Review · Reviewer_YPLo · 2025-10-26

**Soundness:** 3
**Presentation:** 4
**Contribution:** 2
**Rating:** 4
**Confidence:** 3

**Summary:**

The paper design rewards for temporal reasoning to do RL, and experiment on 2 models: Qwen2.5VL 7B and 3B, which beats the baselines. The rewards they design are segment matching rewards and timestamp reward.

**Strengths:**

The paper is clearly written
The reward design is interesting

**Weaknesses:**

1. it is clear that adding temporal related reward will improve the temporal related tasks, it will be cool to test on non-temporal video tasks
2. the novelty of the paper is only on the reward design for temporal tasks, everything else is standard
3. the RL training require a lot of hyperparameter tuning and manual engineering (e.g. 900 steps where first 400 with timestamp reward and 500 without)

**Questions:**

see weaknesses.

---

> ### Author Response · Authors · 2025-11-23
> **Responses**
>
> We sincerely appreciate your thoughtful comments. Our responses are as follows.
>
> **Weakness 1: Generalizability of Our Method**
>
> Please refer to Common Concern 2. Thank you.
>
> **Weakness 2: Method Novelty**
>
> We would like to emphasize the novelty of our approach. Our research focuses on a critical issue in video temporal understanding: existing models often struggle to capture fine-grained, multi-segment temporal information, and to reason over temporal cues effectively. And we made comprehensive design from both reward design (segment matching reward and timestamp reward) and training recipe (phased training recipe). Results in Section 5.5 (Case Study) and Section 6 (Analyses) further illustrate that our novel designs achieve desired outcomes.
>
> **Weakness 3: Engineering Efforts**
>
> Thank you for raising this concern. We would like to clarify the distinction between our approach and straightforward engineering adjustments.
>
> Regarding hyperparameter search, we train our model on only a small portion of the data (10%) and evaluate it on a subset of 200 cases. The reported results are not obtained through cherry-picking among multiple settings on the test sets.
>
> Concerning the design of the phased rewards, the goal is to guide the model to first establish a stable, temporally grounded reasoning process and then allow it to explore more flexible reasoning forms. This design is motivated by our observation that reasoning patterns tend to stabilize after around 400 steps, after which performance gains diminish. The effectiveness and rationale of this approach are further discussed in Section 6.2. Notably, the phased reward design does not originate from ad-hoc engineering trials.

---

### Official Review · Reviewer_ogyi · 2025-11-01

**Soundness:** 3
**Presentation:** 3
**Contribution:** 3
**Rating:** 4
**Confidence:** 5

**Summary:**

The paper proposes MUSEG, a RL-based approach to enhance temporal understanding by introducing timestamp-aware multi-segment grounding. It designs a RL training recipe with phased rewards that progressively guides the model toward temporally grounded reasoning. The experimental results show superior performance over previous methods.

**Strengths:**

1. The paper addresses an important problem and the paper is clearly written. Motivation is clear.
2. Evaluations on in- and cross-domain tasks show good results over previous methods.

**Weaknesses:**

1. To me, the segment matching reward and timestamp reward are likely targeting the same thing. And the reward formats are duplicating. I am wondering the effect of each component. I think there should be an ablation study on how each type of rewards matters to the performance.
2. Since there are several video temporal grounding tasks in the literature, it would be beneficial if authors also evaluate on Ego4D-NLQ, TaCoS, ANet-Captions, QVHighlights.
3. The paper does not have sufficient ablation studies.

**Questions:**

1. How do you handle long and short videos in this setting?
2. How do you make sure the timestamps and video frames are aligned to each other?
3. Could you report results with only phase 1 training?

---

> ### Author Response · Authors · 2025-11-23
> **Responses**
>
> Thank you for your thoughtful comments and questions. Our responses are provided below.
>
> **Weakness 1: Ablation Studies**
>
> We conduct ablation experiments by removing the segment-matching reward, the timestamp reward, and the phased training recipe. As shown in Sections 6.1 and 6.2, removing any of these components leads to a substantial performance degradation, demonstrating that each component is essential to MUSEG. Please let us know if you have any further concerns regarding ablation studies.
>
> **Weakness 2: Evaluation Across More Datasets**
>
> Please refer to Common Concern 1. Thank you.
>
> **Question 1: Handling Videos of Varying Lengths**
>
> Please refer to Common Concern 3. Thank you.
>
> **Question 2: How Timestamps and Video Frames Are Aligned**
>
> In our base model Qwen2.5-VL, timestamps corresponding to video frames are included in the input, enabling the model to align temporal information with visual frames. Our architecture follows Qwen2.5-VL without modification.
>
> **Question 3: Results with Only Phase 1 Training**
>
> Thank you for the suggestion. We provide the results in the table below. We observe that Phase 1 alone improves performance, although the gains are smaller compared with those achieved after Phase 2 training.
>
> | Model | Charades-STA | THUMOS14 | THUMOS15 | E.T. Bench (REF) | E.T. Bench (GND) | E.T. Bench (CAP) | E.T. Bench (COM) | E.T. Bench (AVG) |
> |---|:---:|:---:|:---:|:---:|:---:|:---:|:---:|:---:|
> | Qwen2.5-VL | 50.2 | 24.9 | 23.4 | 51.0 | 30.3 | 16.5 | 9.3 | 26.8 |
> | VideoChat-R1-Thinking | 59.4 | 14.3 | 13.4 | 47.0 | 35.9 | 24.1 | 12.5 | 29.9 |
> | MUSEG-7B (Phase 1) | 57.0 | 27.7 | 26.6 | 59.1 | 37.4 | 23.8 | **19.9** | 35.1 |
> | MUSEG-7B | **59.7** | **29.7** | **29.3** | **60.8** | **38.8** | **25.1** | 19.0 | **35.9** |
>
> * We utilize the official subset of 470 samples from E.T. Bench for our evaluation.

---

### Author Response · Authors · 2025-11-23
**Responses to Common Concerns**

**Common Concern 1: Evaluation Across More Datasets**

Following insightful suggestions from Reviewer ogyi and Reviewer 5XGo, we further report results on more datasets. As shown in the table below, we observe improvements in both in-domain and out-of-domain settings, with larger gains in-domain. This is likely because the OOD datasets mainly involve multiple-choice or captioning tasks that do not require fine-grained timestamp-specific temporal reasoning, unlike grounding datasets and E.T. Bench, which limits achievable improvements of our MUSEG-7B.

| Model | ANet-Captions (ID) | QVHighlights (ID) | AVG (ID) | TemporalBench (OOD) | TempCompass (OOD) | Vinoground (OOD) | AVG (OOD) |
|---|:---:|:---:|:---:|:---:|:---:|:---:|:---:|
| Qwen2.5-VL-7B-Instruct | 31.0 | 19.8 | 25.4 | **67.5** | 71.4 | 62.7 | 67.2 |
| VideoChat-R1-Thinking | 39.0 | 49.1 | 44.1 | 65.6 | 72.5 | 63.0 | 67.0 |
| MUSEG-7B | **48.1** | **61.1** | **54.6** | 67.1 | **72.8** | **63.6** | **67.8** |

*We are unable to report Ego4D-NLQ results because our team could not access the Ego4D videos despite multiple attempts, and this issue appears unresolved for many users. We will include the results once access becomes available. And for TaCoS, since it is included in the training data of both E.T.-Instruct 164K and MUSEG-7B, and no official train / test split is available to our knowledge. We consider it inappropriate to compare model performance on this dataset.

**Common Concern 2: Generalizability of Our Method**

We thank Reviewer YPLo and Reviewer d1dA for raising this concern. We acknowledge that our method brings limited gains on general video understanding tasks, as it primarily enhances timestamp-specific reasoning, which is not required for some questions of general tasks such as “What is the main idea of the video?”.

Moreover, we believe our approach retains broad applicability and remains compatible with complementary techniques aimed at enhancing other aspects of video understanding, as we observe no degradation in general capabilities.

**Common Concern 3: Handling Videos of Varying Lengths**

We thank Reviewer ogyi and Reviewer d1dA for raising this question. We acknowledge that current MLLMs may struggle with both very long and very short videos. For example, for our base model Qwen2.5-VL, its average frame sampling can be suboptimal for long videos, and using “seconds” as the temporal unit can yield less accurate temporal representations in short videos.

Nevertheless, our method shows consistent performance improvement across video lengths, as illustrated in the table below. On THUMOS15, after dividing videos into three equal-sized subsets by duration, we observe substantial improvements on all subsets. On Video-MME, following the official short / medium / long split, we observe no performance degradation in any category, as also shown in Appendix E.

| Model | THUMOS15-short (ID) | THUMOS15-medium (ID) | THUMOS15-long (ID) | THUMOS15 AVG (ID) | Video-MME-short (OOD) | Video-MME-medium (OOD) | Video-MME-long (OOD) | Video-MME AVG (OOD) |
|---|:---:|:---:|:---:|:---:|:---:|:---:|:---:|:---:|
| Qwen2.5-VL-7B-Instruct | 32.9 | 20.8 | 16.7 | 23.4 | 71.8 | 62.7 | 52.6 | 62.4 |
| VideoChat-R1-Thinking | 26.8 | 9.5 | 5.7 | 13.4 | 70.4 | 59.9 | 50.1 | 60.1 |
| MUSEG-7B | **42.3** | **23.7** | **22.1** | **29.3** | **71.9** | **63.7** | **53.6** | **63.1** |

---

### Author Response · Authors · 2025-12-03
**Summary of Reviewer Comments and Author-Reviewer Discussions**

We sincerely thank the reviewers for their valuable comments, constructive suggestions, and the in-depth discussions.

We appreciate that the reviewers acknowledged the contributions of our work:

 - Providing clear motivation and preliminary analyses (reviewers ogyi, d1dA, and 5XGo).
 - Proposing a sound and novel reward design and training recipe for the important problem of temporal understanding in MLLMs (reviewers ogyi, YPLo and 5XGo).
 - Demonstrating solid performance gains on 3B and 7B models across diverse benchmarks (reviewers ogyi, d1dA, and 5XGo).
 - Conducting comprehensive ablation studies (reviewers d1dA and 5XGo).

We also provide a summary of the key concerns and our responses:

**Core Concern 1: Generalization of our method.**

Following suggestions from reviewers ogyi, d1dA and 5XGo, we added experiments on more datasets and video lengths. The results show that our method **generalizes well to other time-sensitive tasks, does not harm overall performance**, and remains compatible with techniques targeting other aspects of video understanding. In addition, our method is consistently effective across videos of **varying lengths**.

**Core Concern 2: Rationale and analysis of our design.**

Thanks to suggestions from reviewers d1dA and 5XGo, we conducted additional experiments and analyses to clarify the motivations behind our design and distinguish it from prior approaches. Together with the ablation studies in our paper, these new results **further validate the design principles and underlying rationale of our method, clearly setting it apart from mere engineering efforts**.

We believe these additions address the reviewers' concerns. We appreciate that **reviewers d1dA and 5XGo acknowledged our clarifications and expressed willingness to raise their scores**. Overall, our work offers a novel and practical approach—both in reward design and training recipe—for enhancing temporal understanding in MLLMs.

---

### Meta-Review · Area_Chair_EDUB · 2025-12-23

**Summary:**

The major concerns that influenced the decision include:  (i) whether the reward components and phased training recipe are essential, (ii) breadth of evaluation and generalization beyond the chosen datasets, and (iii) contribution significance and novelty relative to a closely related contemporaneous work on video temporal grounding (UniTime). During the rebuttal, the authors provided additional experiments, ablations, and a direct comparison to UniTime that substantially reduced these concerns for multiple reviewers. Nonetheless, the core contribution still reads as incremental (primarily reward shaping and a phased schedule), with unresolved clarity on why the reward terms are non-duplicative and limited impact beyond timestamp-centric evaluation.

**Reviewer Concerns:**

Concerns addressed:
1. Necessity of reward components and phased recipe: The authors pointed to ablations removing segment-matching reward, timestamp reward, and phased training, reporting meaningful degradation when a component is removed, which addresses the “duplicative rewards / insufficient ablation” concern. The authors also ran an alternative “partial credit” timestamp reward variant, showed it underperforms, and provided a concrete formulation for partial credit.

2. Evaluation breadth and generalization: The authors added results on additional datasets and provided analyses across video lengths, showing that the method improves timestamp-sensitive tasks without harming general capability.

3. Novelty and comparison vs UniTime: This was a key issue. In follow-up, the authors provided a direct comparison table to UniTime variants (including R1@.5/.7 for fair comparison on single-segment settings) and argued MUSEG is particularly strong on multi-segment grounding where training data is not source-identical; this resolved the reviewer’s request for direct comparison and clearer differentiation.


Concerns Remained:
1. Significance and Novelty: The paper demonstrates clear empirical improvements for timestamp-specific temporal reasoning, but its broader significance is narrower: the main novelty is in reward shaping and a phased training schedule. As such, some reviewers may still view the work as incremental, with contributions that can be limited in conceptual scope.

2.  Redundancy between segment-matching and timestamp rewards: The authors only show that removing each reward decreases the performance, yet they didn't explain why the two rewards are not functionally duplicative, or provide a clear decomposition showing what failure modes each reward uniquely addresses.

2. Generalization to non-temporal video understanding tasks: The authors argue that limited gains on generic video capabilities are expected because the method targets timestamp-specific reasoning, and they show no degradation. However, the method’s broader impact beyond timestamp-centric tasks remains limited by design.

**Reviewer Scores:**

ogyi: Started at 4. Since the redundancy/ablation concern is only partly addressed, they’d likely stay at 4.

YPLo: Started at 4. Novelty and non-temporal generalization are still open, so they’d likely stay at 4.

d1dA & 5XGo: Both said in discussion they would raise to 6.

---

### Decision · Program_Chairs · 2026-01-26

Reject